# A study of Chinese enterprises' business models to determine the impact of dynamic capabilities on innovation and performance

Yan Jingwen[1], Azmawani Abd Rahman[1]*, Tong Tong[2], Nitty Hirawaty Kamarulzaman[3], Shafie Bin Sidek[1]

1 School of Business and Economics, Universiti Putra Malaysia, Serdang, Selangor, Malaysia, 2 Department of Literature, Sichuan Minzu College, Kangding, China, 3 Department of Agribusiness and Bioresource Economics, Faculty of Agriculture, Universiti Putra Malaysia, Serdang, Selangor

* azar@upm.edu.my

**Data Availability Statement:** All relevant data are within the manuscript and its Supporting Information files.

## Abstract

Small and medium-sized enterprises (SMEs) can gain a competitive advantage by implementing business model innovation (BMI), which is characterized as irreversible changes to a company's business model. However, BMI is often associated with high risk, uncertainty, and ambiguity. In this study, the effectiveness of BMI on improving SME performance is examined using structural equation modeling (SEM) based on data collected from 330 Chinese SMEs. The purpose of this paper is to examine how enterprise risk management (ERM), organizational agility (OA), and entrepreneurial orientation (EO) affect SME performance. The results reveal that ERM, OA, and EO all have a positive impact on efficiency-centered BMI and SME performance; efficiency-centered BMI mediates this pathway. Building on dynamic capabilities theory, this paper combines ERM, OA, and EO into one framework to assess their impact on SME performance. Additionally, a case study is presented to provide suggestions for making decisions about BMI implementation.

## Introduction

New competitors do not always have to be large, established players [1, 2] but can also be start-ups that offer another business model to that of the incumbent [3–5]. With the introduction of new business models, the game's rules change significantly in specific industries [6]. It is imperative that incumbent companies continually innovate and transform their business models to maintain their market position [7]. Business model innovation (BMI) involves significant changes in the company's infrastructure or business model [8]. BMI helps companies maintain their competitive advantage [9] by generating activities beyond product and process innovation [10].

Researchers have been examining factors that have contributed to advancing firm BMI in recent years by facilitating firms' capacity to respond to environmental changes [11–13]. BMI may require companies to cannibalize their current revenue streams to generate uncertain future revenues [11, 14, 15]. Thus, it is often difficult to predict how long, when, and whether

**Funding:** The author(s) received no specific funding for this work.

**Competing interests:** The authors have declared that no competing interests exist.

BMIs will be successful [16]. Due to their ample resources, large companies can usually experiment with new business models [17]. Furthermore, new prototype business models can be developed as spinoffs without posing a threat to the company's survival in large companies [18]. Small and medium-sized enterprises (SMEs) are less likely to experiment with new business models because they have limited resources, and it is also challenging for SMEs to cope with the risks and challenges of BMIs if they fail [19].

Nevertheless, SMEs can minimize uncertainty by continuously identifying upcoming opportunities and threats within and outside their borders, leveraging these, and making informed decisions [20]. Dynamic capability theory states that BMI allows SMEs to adapt to market changes and environmental shifts more rapidly. SMEs that continuously innovate their business models are more resilient to economic downturns and industry disruptions, thereby ensuring sustained performance [17]. BMI positively impacts SMEs' performance by enhancing their innovation capabilities and market responsiveness, leading to better financial outcomes and competitive positioning [21]. SMEs committed to BMI are better positioned to leverage organizational learning, which enhances their innovative capabilities and leads to long-term performance improvements [22].

However, the failure of BMI in SMEs can have a catastrophic impact [23], rendering effective risk management (ERM) crucial. ERM is a systematic approach to risk management that encompasses the entire organization. It involves identifying, assessing, and responding to opportunities and threats that may impact the organization's objectives (Institute of Internal Auditors, 2009). By adopting an enterprise-wide approach to risk management, organizations can promote risk awareness and understanding throughout the corporate structure, tapping into the knowledge and suggestions of people at all firm levels [24]. This allows organizations to better manage risks, respond to changes in the market environment, and ensure sustainable growth and development [25].

Studies have indicated that ERM significantly impacts BMI by enhancing risk identification, strategic alignment, decision-making, and resource allocation. By integrating ERM with their innovation processes, organizations can effectively manage the risks associated with new business models, ensuring sustained innovation and improved performance [26–28]. Effective ERM practices help organizations identify not only potential risks but also opportunities for innovation. According to Mikes and Kaplan (2014) [27], ERM can uncover market gaps and emerging trends that can be addressed through innovative business models. ERM ensures that risk management strategies are aligned with the organization's strategic goals, which is crucial for a successful BMI. Bromiley et al. (2015) [28] indicate that firms with mature ERM systems are more agile and capable of adapting their business models to align with strategic changes. Barton et al. (2002) [29]emphasize that ERM provides a structured framework for strategic planning, enabling firms to pivot their BMI in response to evolving risks and opportunities. While ERM has the potential to improve organizational performance through better risk management, strategic decision-making, and enhanced resilience, its effectiveness is not guaranteed. Aebi et al. (2012) [30] suggest that for some organizations, especially smaller firms, the costs associated with ERM might outweigh the benefits as ERM may have negligible or even negative impacts on performance. Bromiley et al. (2015) [28] note that poorly executed ERM initiatives may not yield the expected benefits and can sometimes become bureaucratic exercises that add little value.

According to past research, the impact of ERM on performance is not guaranteed. A prominent feature of China's SMEs is bureaucracy [31]. Thus, although ERM may promote BMI, the relationship between ERM and SMEs' performance in China is complicated due to the need for cost control and concerns about BMI failure. Syrová and Špička (2023) [32] suggest that the connection between ERM and firm performance is not straightforward but is influenced

by various mediating factors. Some notable mediators include BMI [33] and strategic agility [34]. Research in this field has primarily focused on publicly listed companies and major international corporations, while studies on SMEs have been relatively limited [34–37].

Organizational agility (OA) is a frequently considered factor for improving business performance in strategic management as agile SMEs can respond quickly to market demand and environmental changes, enhancing their competitive advantage. Research has determined that businesses should have the ability to swiftly perceive opportunities and innovate products or services to meet customer needs and improve business performance [38, 39]. Organizations need to respond quickly to external changes and seize opportunities to adjust their business operations to promote improved corporate performance [40]. Hence, OA has a positive impact on performance.

Moreover, China has been actively implementing a "mass innovation, mass entrepreneurship" policy since 2014. This policy is oriented around promoting entrepreneurship. According to the data, there were 844,000 newly registered firms in the first quarter of 2015, which was a 38.4% increase from the previous year. The newly registered capital amounted to 4.8 trillion yuan, reflecting a significant increase of 90.6% and underscoring the significant impact of "mass innovation, mass entrepreneurship" policy on the corporate social responsibility undertaken by SMEs [41]. Hence, it is imperative to include entrepreneurial orientation (EO) as a crucial factor when examining the performance of SMEs in China.

EO is a strategic orientation representing a behavioral model for firms regarding innovation, risk-taking, and initiative [40, 41]. Cope (2005) [42, 43] argued that entrepreneurial orientation allows firms to gain a competitive advantage and thus improve their financial performance. Another study found that both market orientation and entrepreneurial orientation have positive performance effects in emerging markets [44]. Several recent studies have determined that EO is beneficial to SMEs because it enables them to move in the direction of innovation and to proactively utilize resources and take risks to benefit their performance [45].

In summary, ERM, OA, and EO each play a critical role in enhancing SME performance. ERM helps manage risks and align strategies, OA provides the flexibility to adapt to changes, and EO fosters innovation and proactivity. Together, these elements create a dynamic and resilient organizational environment, enabling SMEs to thrive in competitive and uncertain markets. However, some studies have shown that the relationship between ERM and SME performance is not significant [44, 45]. The expenses associated with BMI significantly contribute to SMEs' failure, and the influence of EO on performance is only considerable in the long run, with minimal improvements in short-term performance [46, 47]. ERM, OA, and EO have generally been examined in isolation, with few studies placing them within the same framework. Additionally, due to differences in research backgrounds, the results of these studies are also contradictory.

In the present study, the primary objective is to examine the antecedents of BMI and their impact on SMEs' performance. Specifically, this study examines how BMI influences SMEs' performance directly and indirectly. It also addresses how BMI mediates the relationship between SMEs' ERM, OA, and EO in line with dynamic capabilities theory. SMEs' owners and managers are provided with helpful management ideas and practical suggestions for improving performance and reducing risks. The following research questions are examined:

- Does BMI influence SMEs' performance in China?

- What is the connection between ERM, OA, and EO and SMEs' performance?

The paper proceeds as follows. In the literature review, multiple hypotheses and a model are proposed. The methodology chapter explains the data collection procedures and analytical

techniques used to test the hypotheses and evaluate the model. The results are reported in the next chapter, followed by the conclusion, where practical suggestions and limitations are outlined.

## An overview of the literature and hypotheses

### Enterprise risk management (ERM)

The Committee of Sponsoring Organizations of the Treadway Commission (COSO) released the ERM Framework in 2004. By 2005, the COSO framework had gained significant traction, becoming a foundational tool for ERM implementation globally [48]. In 2009, the International Organization for Standardization (ISO) published ISO 31000, offering principles and generic guidelines on risk management. Unlike the COSO's framework, which is more detailed, ISO 31000 provides a high-level overview of ERM applicable across different industries [49]. The financial crisis of 2007–2008 highlighted the need for robust risk management. Regulatory bodies worldwide, such as the U.S. Securities and Exchange Commission (SEC) and the Basel Committee on Banking Supervision, began to emphasize ERM in corporate governance and financial regulation [50]. Organizations began integrating ERM more deeply into their strategic planning processes. This integration was driven by the recognition that effective risk management could provide a competitive advantage and improve strategic decision-making [51].

Beasley et al. (2017) developed ERM maturity models to assess the sophistication of ERM practices within organizations. These models helped firms benchmark their ERM practices and identify areas for improvement [52]. The rise of big data, artificial intelligence (AI), and machine learning brought significant changes to ERM. These technologies enabled more precise risk modelling, predictive analytics, and real-time risk monitoring [53]. There was a growing emphasis on cultivating a risk-aware culture within organizations. Leadership and corporate governance structures play a critical role in embedding ERM into the organizational culture [54]. Such as environmental and social risks, into ERM frameworks given the importance of these risks for long-term resilience and stakeholder trust [55].

Baxter et al. (2013) [56] concluded that companies with robust ERM practices tend to have higher market valuations and better credit ratings. They maintain that ERM provides a structured approach to managing risks, which reassures investors and reduces the cost of capital. Florio and Leoni (2017) [57] examined the relationship between ERM maturity and financial performance in European companies; they found that firms with more mature ERM processes exhibited superior financial performance, indicating that the depth and integration of ERM processes play a critical role in achieving financial benefits. Brustbauer (2016) [58] highlighted that ERM supports strategic risk management by integrating risk considerations into strategic planning processes. This alignment helps organizations anticipate and mitigate potential strategic risks, thereby enhancing decision-making and long-term planning. Farrell and Gallagher (2015) [26] studied the impact of ERM on competitive advantage and concluded that firms with comprehensive ERM frameworks are better positioned to navigate uncertainties and capitalize on opportunities. Their findings suggest that ERM can be a source of competitive differentiation by enhancing agility and responsiveness.

Gates, Nicolas, and Walker (2012) [59] demonstrated that ERM contributes to operational efficiency by promoting risk awareness and encouraging proactive risk management practices. Their research indicated that organizations with well-implemented ERM frameworks tend to have more efficient operations and better resource allocation. Capano et al. (2020) [60] explored the role of ERM in enhancing organizational resilience and determined that companies with advanced ERM practices were better able to withstand and recover from crises, such as the COVID-19 pandemic. ERM practices provided a structured approach to crisis

management and facilitated a quicker recovery. Lechner and Gatzert (2018) [61] found that as organizations become more complex, the challenges of implementing effective ERM practices increase. This complexity requires more sophisticated risk management approaches and greater coordination across the organization.

## Organizational agility (OA)

Technology advancements drive enterprises toward intelligent development and digital transformation in the 21st century, and enterprise business models and industry environments—which were once the norm for companies—are changing dramatically in response to changing times and technology. An organization's ability to respond continuously and rapidly to unpredictable changes in customer needs is a crucial characteristic of organizational agility in a changing and unpredictable business environment [62]. As a result of continuously changing business environments and high competition pressure, organizations need to be capable of adjusting to agility in the ever-changing industry situation [63]. OA occurs when an entity puts itself in the customers' shoes and provides higher quality, less costly [64]. OA is also essential to respond quickly to changes in the internal and external environment and provide high-quality products and services at a reduced cost to the company.

OA has emerged as a vital capability for firms operating in volatile, uncertain, complex, and ambiguous (VUCA) environments. The rise of digital technologies has played a critical role in enhancing OA, with organizations leveraging big data, cloud computing, and artificial intelligence to improve their responsiveness and decision-making capabilities [65]. Teece's (2007) [66] combined dynamic capabilities framework links OA to firms' ability to integrate, build, and reconfigure internal and external competencies to address rapidly changing environments. This framework highlights three main capabilities: sensing opportunities and threats, seizing opportunities, and transforming operations.

Doz and Kosonen (2010) [67] introduced the concept of strategic agility, which focuses on organizations' ability to continuously adjust their strategic direction and operations in response to changing environments. Using a case study model of organizational agility, Crocitto et al. (2003) [68] concluded that organizations were better able to react to sudden environmental changes when they were more agile, thereby enhancing organizational resilience and enabling firms to withstand and recover from disruptions such as economic crises and technological changes [69].

The COVID-19 pandemic underscored the importance of agility as organizations that could pivot quickly were better able to navigate the uncertainties and challenges of the crisis. Agile organizations are better positioned to drive innovation both in terms of product development and process improvements. This continuous innovation cycle is crucial for maintaining relevance and customer satisfaction in dynamic markets [70]. By responding quickly to customer needs and feedback, agile firms can enhance customer loyalty and market responsiveness, further solidifying their market position [71]. Organizations with high agility are able to innovate more effectively, bringing new products and services to market faster than their less agile competitors [72]. Adopting agile methodologies across various business functions can improve process efficiency and flexibility, and continuous process innovation ensures that operations remain responsive to changing demands [73].

## Entrepreneurial orientation (EO)

According to Miller (1991) [74], EO has three meanings, one is a risk-facing behavior, the other is an innovation-embracing behavior, and the third is an ability to react tendency oriented policy before competitors. Lumpkin and Dess (1996) [75] first introduced and defined

entrepreneurial orientation as behavior that facilitates the creation and expansion of businesses. A firm's entrepreneurial orientation can be defined as a particular set of behaviors and processes related to its entrepreneurial activities, including formulating and implementing strategies, promoting product and service development, and exploiting market opportunities [76]. According to Stone and Good (2004) [77], entrepreneurial orientation is a way for entrepreneurs to promote new development while developing new ventures.

Per Lumpkin and Dess (1996) [75], risk refers to the possibility and probability of losses or negative results in financial analysis. Different applications of risk have different meanings, and current applications cover a wide range of types. According to entrepreneurship theory and strategic management theory, risk-taking is an entrepreneurial orientation characterized by firms' tendency to act boldly and make quick decisions when the outcome of their entrepreneurial activities is unknown. Entrepreneurial orientation involves taking risks, such as venturing into unexplored areas or markets, investing capital into uncertain projects, and taking out loans [75]. By investing in high-risk projects, a firm avoids relying only on tried-and-true activities and incorporates risks into its business model [78]. Dess and Lumpkin (2005) [75] present an integrated discussion of risk-taking. They define risk-taking as firms' tolerance for risk and uncertainty and the extent to which they are willing to take advantage of risky opportunities. To achieve high profits, new firms with a higher risk tolerance choose more difficult projects during business development [79].

According to Mishra (2017) [80], a company's competitive advantage is derived from the value creation and adaptation mechanisms of its entrepreneurial orientation, which combines resource and positioning logic. For Mishra, entrepreneurship can influence a firm's performance and core competitiveness in two ways: an organization's adaptability can be enhanced as a result of generating core resources, and value activities can be expanded by tapping into a wide range of resources. This logic emphasizes a long-term development process concerning core resource generation through enhanced learning, but a firm must first survive, which is a challenge for startups. As a result, the orientation logic of value activity formation through adaptive mechanisms is more conducive to the short-term survival challenges startups face [80].

EO contributes to strategic renewal by fostering an entrepreneurial mindset that encourages continuous re-evaluation and adaptation of business strategies [81]. This ongoing strategic renewal is essential for firms operating in rapidly changing environments [82]. Firms with high EO tend to perform better financially due to their ability to innovate, take calculated risks, and proactively seize market opportunities [83, 84]. Firms with high levels of EO are also more likely to develop new products and services, engage in research and development(R&D) activities, and adopt new technologies [84]. The ability to innovate is a critical driver of long-term competitive advantage and market success [85]. EO enhances market performance by improving customer satisfaction, brand recognition, and market share [86]. Proactive and innovative behaviors allow firms to better anticipate and meet customer needs, leading to stronger market positions [87].

## Business model innovation (BMI)

The interest in BMI among academics has grown significantly in recent years [88]. BMI involves establishing a fundamentally new business model within an existing company [89] or defining a new business logic for the firm and a new way to maximize stakeholder value [90]. A similar increase has been observed in business models and BMI-related academic conferences and management workshops. The BMI concept, however, is tricky to evaluate [90]. BMI is often divided into novelty-centered BMI and efficiency-centered BMI to better understand the different strategic approaches organizations can take to create and capture value. These

two categories emphasize distinct mechanisms for achieving competitive advantage and are driven by different innovation objectives. Novelty-centered BMI focuses on introducing new elements or recombining existing elements in a unique way to create unprecedented value propositions, revenue models, and market offerings [91]. Efficiency-centered BMI aims to enhance the efficiency of existing business models by optimizing processes, reducing costs, and improving resource utilization [17].

By distinguishing between novelty and efficiency, organizations can better align their innovation strategies with their overall business goals and competitive environments [92]. Novelty-centered BMI often targets differentiation and market disruption, while efficiency-centered BMI aims at cost leadership and operational excellence. Additionally, novelty-centered BMI may demand higher investment in R&D and creative processes, whereas efficiency-centered BMI focuses on optimizing existing resources and capabilities [93]. This differentiation helps businesses understand the distinct pathways to achieving competitive advantage through either disruptive new offerings or optimized, efficient operations. Table 1 compiles the impact of two types of BMI on performance.

## Hypotheses

**Relationship between enterprise risk management (ERM) and SMEs' performance.**
ERM has emerged as a critical framework for identifying, assessing, and managing risks in organizations. For SMEs, implementing ERM can significantly influence performance outcomes due to their limited resources and higher vulnerability to risks compared to larger firms. Effective risk identification and assessment allow SMEs to anticipate potential threats and opportunities, leading to better strategic decision-making and resource allocation. Studies indicate that SMEs that excel in these initial ERM stages can mitigate adverse effects and exploit new opportunities, enhancing overall performance [102]. Bromiley et al. (2015) [28] discovered that risk identification and assessment are positively correlated with improved strategic outcomes in SMEs as these processes enable firms to prioritize risks and align their

**Table 1. The impact of novelty-centered and efficiency-centered BMI on performance.**

| Scholars | Independent variable | Mediating variables | Moderating variables | Dependent variable | Findings |
|---|---|---|---|---|---|
| Zott & Amit (2002) [94] | novelty-centered, efficiency-centered | | | firm and market performance | positive correlation |
| Zott & Amit (2007 [4]) | novelty-centered, efficiency-centered | | external environment | firm performance | complex correlation, positive moderation |
| Zott & Amit (2008) [91] | novelty-centered, efficiency-centered | product market strategy | | firm performance | positive correlation, mediation |
| Rhoads (2011) [95] | novelty-centered, efficiency-centered | | | firm performance | positive correlation |
| Brannon (2011) [96] | novelty-centered, efficiency-centered | customer information processing | organizational inertia | firm performance | positive correlation, mediation, negative moderation |
| Brettel (2012) [97] | novelty-centered, efficiency-centered | | relationship marketing efforts | firm performance | positive correlation, negative moderation |
| Pati Rakesh (2018) [98] | novelty-centered, efficiency-centered | | business age and external environment | firm performance | positive correlation, complex moderating. |
| Anwar (2018) [99] | novelty-centered, efficiency-centered | competitive advantages | | firm performance | positive correlation, mediation |
| Balboni (2019) [100] | novelty-centered, efficiency-centered | | | firm performance | complex correlation |
| Nancy M.P et al. (2019) [101] | novelty-centered, efficiency-centered | dynamic capabilities | | firm performance | positive correlation, mediation |

strategies accordingly. Hoyt and Liebenberg (2011) [103] demonstrated that SMEs with well-defined risk response mechanisms experience less volatility in performance metrics, suggesting that ERM contributes to a more stable and predictable business environment. Frigo and Anderson (2011) [54] revealed that SMEs that integrate ERM into their strategic planning processes achieve better alignment between risk management and business goals, resulting in improved operational efficiency and market performance.

In addition, several studies have found that enterprises perform better with an effective ERM system [105]. As well as evaluating performance indicators for public corporations, ERM tools can also be used to assess indicators for private corporations [104]. The ability to manage enterprise risks effectively can reduce the probability of their occurrence and the degree of their impact, reduce the negative impact of uncertainty on enterprises, and improve their competitiveness and stability. Businesses can better identify, assess, and respond to risks by improving ERM capacity. Therefore, by strengthening risk management capabilities, enterprises can better cope with risks in different business environments and thus improve their performance. Based on these points, the following hypothesis is formulated:

**Hypothesis 1.** ERM positively influences SMEs' performance.

**Relationship between organizational agility (OA) and SMEs' performance.** In strategic management, how to obtain and maintain a company's inherent competitive advantage in the face of a constantly changing environment and increased uncertainty and risk in the business operations process is a significant challenge [105, 106]. Roberts and Grover (2012) [70] argue that firms should have the ability to quickly perceive opportunities and innovate products or services to respond to customer needs, which then facilitates the improvement of firm performance. Yang and Liu et al. (2020) [71] support that organizations need to respond quickly to external changes and seize opportunities to adjust business operating procedures and improve firm performance. Therefore, it is believed that organizational agility positively impacts business performance.

Côrte-Real et al. (2017) [40] argue that agility directly leads to better performance at the process level and in terms of competitive advantage. They also suggest that organizations invest in big data tools to support agility, which may ultimately lead to sustainable competitive advantage. Sambamurthy et al. (2003) [69] emphasize that business process agility benefits enterprise value realization because it helps achieve superior financial performance. Companies that also have effective business analysis procedures can adjust their business processes promptly, develop processes that are more responsive to customer and market needs, take proactive actions to retain customers, respond to customer needs, and improve operational flexibility; as a result, companies can increase revenue while reducing costs [96, 97].

**Hypothesis 2.** Organizational agility (OA) positively influences SMEs' performance.

**Relationship between entrepreneurial orientation (EO) and SMEs' performance.** Scholars have conducted numerous empirical studies on the contribution of EO to performance. Peters and Waterman (1982) [107] argue that entrepreneurial activities contribute to firms' financial performance, while Cope (2005) [108] argues that EO allows firms to gain a competitive advantage and, thus, improved financial performance. Miller and Friesen (1983) [109] argue more specifically that the dimensions of EO, such as innovativeness, and risk-taking, are positively related to firm performance. Many scholars maintain that EO is essential to improving financial performance and thus gaining a competitive advantage [110, 111]. Smart and Conant (1994) [112] observe that high OA improves competitive differentiation. Additionally, Zahra and Covin (2000) [113] conclude that there is a positive relationship between intra-firm entrepreneurial activity and performance. This relationship changes over time, with a more substantial long-term rather than short-term effect. Shane et al. (2001) [114] identify a positive relationship between entrepreneurial innovation and firm performance, and Gruber-

Muecke (2015) [44] note that both market orientation and EO have positive performance effects in emerging markets. Recent studies have shown that EO is beneficial to SMEs by enabling them to move in the direction of innovation and to proactively and riskily utilize resources to benefit their performance [45].

**Hypothesis 3.** EO positively influences SMEs' performance.

**Relationship between efficiency-centered BMI and SME performance.** SMEs are vital to the global economy and are often characterized by their flexibility, innovative potential, and significant contribution to employment and economic growth. BMI has emerged as a crucial factor in enhancing the performance of SMEs by enabling them to adapt to changing market conditions and to leverage new opportunities.

Based on Zott and Amit's (2012) [115] classification of BMI, this study focuses on novelty-centered business models and efficiency-centered business models. Efficiency-centered business models evaluate how a firm can reduce transaction costs (e.g., by lowering transaction costs for each counterparty). The ride-sharing company Uber allows the company to earn a share of revenue without keeping expensive assets on its books, giving it an advantage over its competitors. There is no need for cars, cab licenses, or local street knowledge. The platform, driver contracts, and brand are the most critical assets. As other companies follow suit and compete, Uber's market and strategic advantages will become smaller and smaller as it enters new markets instead of expanding existing ones. According to this study, BMI is associated with firm performance based on the following assumptions. First, rather than focusing on output efficiency per se, efficiency-centered business models aim to improve transactional efficiency (for instance, reducing transaction costs for each counterparty). Reducing transaction costs is the purpose and central tenet of efficiency-centered BMI. The order tracking system in Amazon's business model is an example of an efficiency-based BMI that increases transparency and efficiency. This example also illustrates that business model innovation themes are not mutually exclusive, and Amazon's business model contains both efficiency and novelty components. By avoiding transaction risk and reducing complexity and uncertainty for the firm's counterparties, efficiency-based business models enable lowered transaction costs and improved firm performance [4]. The most essential characteristic of efficient BMI is the sharing of information [91].

Trust between partners minimizes information asymmetry [116], which enables companies and other stakeholders to accurately and quickly grasp market and customer demand information and adjust their products and services accordingly. Second, companies that adopt efficiency-based BMI can achieve a state of real-time management for value systems such as inventory monitoring, transaction accuracy, and product traceability; this is made possible by shortening the processing time of orders and customer demands, which significantly increases partners' dependence on the company and improves the value parties and overall operational efficiency [117]. Companies can also reduce communication and search costs and become less likely to engage in opportunistic behavior by implementing efficient BMI. Furthermore, efficiency-based BMIs can speed up the transaction process by reducing the possibility of errors and simplifying the process. The coordination costs among the various stakeholders in the value chain and the search costs of the other stakeholders are significantly reduced [118].

Finally, efficiency-based BMI improves decision-making efficiency. A more rapid decision-making process, more precise and rich information, and fewer transaction errors will result in more reliable, concise, and smoother transactions, information transfers, and decision-making processes [119]. Transaction costs have a direct impact on corporate profits [120]. Efficient business models innovate to reduce transaction costs to provide better products and services [121]. Successful examples of firms achieving low-cost advantages through efficiency-based BMI abound. For example, Southwest Airlines applied efficiency BMI to beat mainstream

airlines with "point-to-point" flights and limited air service; Nucor, a small steel mill, beat traditional large steel companies through efficiency BMI. Spring Airlines gained a low-cost competitive advantage by innovating its efficient business model. BMI is an effective way for companies to break with the original business model of their industries so that they can compete against high-cost and low-efficiency rivals. Streamlined business models offer cost reductions and efficiency improvements to customers, which in the short term can reduce costs and improve transaction efficiency for all parties and in the long term can improve business performance. Thus, the following hypothesis is proposed:

**Hypothesis 4 (H4).** Efficiency-centered BMI positively influences SMEs' performance.

**Relationship between novelty-centered BMI and SME performance.** Novelty-centered BMI focuses on introducing new elements or recombining existing elements in a unique way to create unprecedented value propositions, revenue models, and market offerings [91]. This type of BMI is characterized by creativity and differentiation, aiming to disrupt existing markets or create entirely new ones. Businesses create new markets and partnerships through novel BMIs to create a new market space [122] and attract more customers [75]. Novelty-centered BMI can break down barriers to products or services in the traditional sense and can capture added value in the original market. For example, Dell replaced its standard retail business model with online customization to gain a competitive advantage [4]. By designing a new marketing concept for enterprises, novelty-centered BMI can offer unique consumer experiences and even open up whole new markets. For example, Disney's business model, which combines culture with technology and multichannel cash, visitors to the park have had a much better experience.

Novelty-centered BMI has brought considerable profits to Apple. Apple's iPod–iTunes Store model resulted in nearly all music downloads in the United States, and this novel BMI has enabled Apple's products to last much longer than similar products. Innovating business models means offering customers products, services, or experiences that are new to them and that stimulate their willingness to buy. Many companies exploit the internet and related technological developments to create new products and services (deal content); these developments are also used to involve stakeholders in new ways within the transaction framework (deal structure and governance), thereby increasing the bargaining power of the company itself [119]. However, though novelty-centered BMI can enhance a company's competitiveness, it may affect its financial metrics over time. BMI often needs to cultivate user habits, and it takes time for enterprises to explore and accumulate experience in the new business model.

The effect of novelty-centered BMI also need time to be tested and confirmed. One example of Jing Dong (JD), a giant technology company in China that is both a platform builder and a provider of merchandising services. JD first achieved full-year profitability in 2016. The key to JD continued recognition by capital markets despite years of failure to achieve profitability is that JD Mall has always adhered to its self-operated model and built its logistics, focusing on cutting down on intermediate links, greatly enhancing the consumer experience, and providing consumers with quality products and services in the shortest possible time. Indeed, evaluating the merits of a novelty-centered BMI does not depend on short-term financial indicators. Shaping a novelty-centered BMI may require long-term accumulation to achieve profitability. The ability to innovate long-term is one of the most essential characteristics of business models. Long-term evaluation is the best way to determine whether a novelty-centered BMI is of a high quality. Many BMIs, which may not be profitable in the short term, are difficult to surpass in terms of profitability once protection mechanisms are established. Hence, the following hypothesis is proposed:

**Hypothesis 5 (H5).** Novelty-Centered BMI positively influences SME performance.

**The mediating role of BMI.**    The business environment changes rapidly, and companies face more risks and uncertainties. In such an environment, firms must have dynamic capabilities to cope with changes and achieve continuous innovation. Based on dynamic capability theory, this thesis explores how ERM, OA, and EO affect BMI and consequently SME performance. Though researchers have considered how firms adapt to environmental changes using the dynamic capabilities framework, studies are still lacking on whether dynamic capabilities remain effective in extreme circumstances such as uncertainty crises and especially changing environments. Nair (2014) examines whether an enterprise's enterprise risk management (ERM) capabilities adequately respond to uncertainty crises by focusing on ERM as a dynamic capability. The results suggest that firms may need dynamic capabilities to react and respond to different dimensions and types of environmental changes [123, 124].

When firms are facing adverse economic conditions, ERM can be critical to reducing risk, maintaining stability, and gaining a competitive advantage. OA is widely recognized as a high-level dynamic capability that can continuously improve business performance by simultaneously and efficiently responding to consumer needs over a long period of time. Enterprise agility facilitates the coincidental alignment and allocation of corporate resources (e.g., assets, knowledge, relationships, etc.). Firms engaging in BMI enhance their ability to perceive potential business opportunities and address challenges. In response to environmental changes, rules and models can be re-established through resource innovation and capability enhancement, thereby allowing for strategic adjustments. OA can manifests in executive execution, managerial action, and strategic decision-making, which depend on these capabilities. EO is essentially a value orientation for companies to pursue innovation, take risks, and act before competitors, stimulating companies' intention to innovate and prompting them to implement more innovative behaviors.

Per the above analysis, ERM, OA, and EO have dynamic capabilities. ERM provides a foundation for identifying and mitigating risks, while OA ensures that SMEs can adapt to changes swiftly. EO drives innovation and proactive market behavior. The combination of these elements enables SMEs to navigate uncertainties, capitalize on opportunities, and maintain competitive advantage. However, some scholars have suggested that dynamic capabilities directly affect performance [112, 114, 115], and Eisenhart and Martin argue that dynamic capabilities are not sufficient to directly improve organizational performance [125]. Winter and Zahra (2006) [126] suggest that dynamic capabilities affect performance and need to be realized through intermediate vectors. Wilden et al. (2013) [127] analyzed the mechanism of dynamic capabilities' impact on performance from the perspective of organizational structure. They found that startups' dynamic capabilities positively affect performance through embedded internal and external structures.

In this context, BMI has gained significant attention in recent years as a critical driver of organizational performance. The literature consistently demonstrates that BMI is a critical driver of organizational performance. By enabling firms to differentiate themselves, adapt to market changes, manage revenue and costs more effectively, and foster a culture of innovation, BMI contributes significantly to enhanced performance [97, 128–130]. The impact of BMI's effect on enterprises is presented in Table 1.

A number of past studies have used BMI as a mediator, and it has been found that BMI mediates the relationship between EO and firm performance by transforming entrepreneurial initiatives into structured and scalable business models. Bouncken and Fredrich (2016) [131] emphasize that innovative business models mediate the positive effects of EO on firm performance. BMI acts as a mediator by converting technological advancements into new business models that drive innovation and market success. Firms with strong technological capabilities often innovate their business models to better utilize new technologies and meet market demands

[132–134]. BMI mediates the impact of strategic orientation on market performance by enabling firms to align their strategic goals with innovative business practices. Strategic orientation drives the need for new business models that can better capture and deliver value [11, 125, 126].

In this research, we also believe BMI plays a mediating role in performance. Dynamic capabilities involve the firm's ability to integrate, build, and reconfigure internal and external competences to address rapidly changing environments. BMI can mediate the relationship between dynamic capabilities and competitive advantage by enabling firms to reconfigure their business models in response to environmental changes. Firms with robust dynamic capabilities use BMI to adapt to market shifts and maintain competitive advantage [6, 17]. Dynamic capabilities facilitate BMI, which in turn enhances competitive advantage, and firms use BMI to adapt their business models in dynamic environments.

In sum, the mediating role of BMI is crucial for translating strategic orientations, entrepreneurial initiatives, and technological capabilities into improved performance outcomes. BMI provides firms with the necessary framework to adapt and reconfigure their business models in response to changing market conditions and technological advancements. Firms that effectively utilize BMI as a mediator are more likely to achieve sustained competitive advantage through continuous innovation and strategic alignment. We therefore propose the following hypotheses. The conceptual model is available in **Fig 1**.

**Hypothesis 6 (H6a).**   Efficiency-centered BMI mediates the relationship between ERM and SME performance.

**Hypothesis 6 (H6b).**   Novelty-centered BMI mediates the relationship between ERM and SME performance.

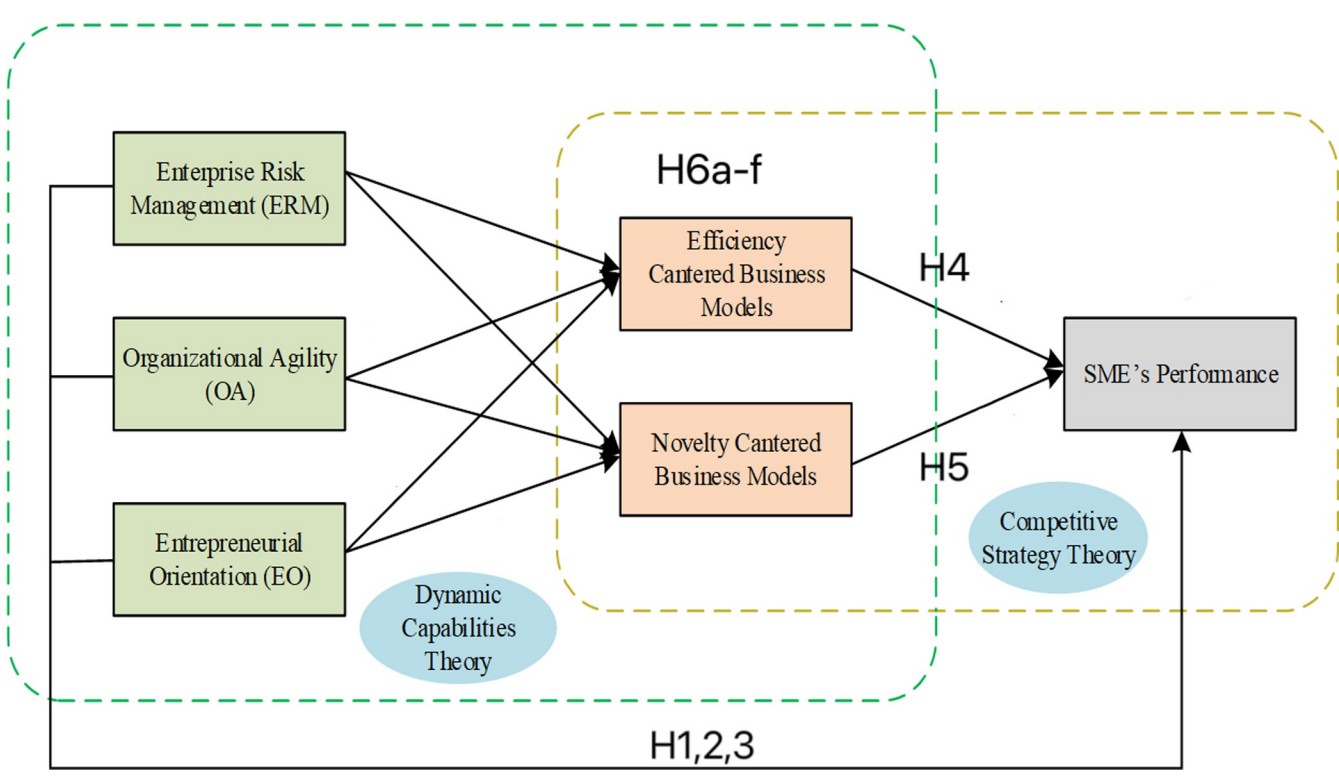

**Fig 1. The conceptual model in the research.**

**Hypothesis 6 (H6c).** Efficiency-centered BMI mediates the relationship between OA and SME performance.

**Hypothesis 6(H6d).** Novelty-centered BMI mediates the relationship between OA and SME performance.

**Hypothesis 6 (H6e).** Efficiency-centered BMI mediates the relationship between EO and SME performance.

**Hypothesis 6 (H6f).** Novelty-centered BMI mediates the relationship between EO and SME performance.

## Methodology and data collection

### Sample and procedure

Questionnaires were used to collect data. The "mass entrepreneurship and mass innovation" policy has allowed China to increase SMEs and accelerate the country's economic growth. According to the National Bureau of Statistics, the top 20 Chinese cities for number of SMEs created nearly $30 trillion inprofits in 2021, accounting for more than 60% of the country's gross domestic product. The top 20 cities are mainly in central China, particularly in the six contiguous provinces of Shanxi, Henan, Anhui, Hubei, Jiangxi, and Hunan. These provinces' income level and labor–capital ratio are approximately equivalent, resulting in comparable economic well-being and standard of living. Consequently, the area can be characterized as a homogenized population area. Owing to limitations in funding and personnel resources, data collection mostly focused on Henan Province (first pilot province of "mass entrepreneurship and mass innovation" policy), which hosts all the top 20 cities with the highest number of SMEs.

The economic development of Henan Province has been rapid since"mass entrepreneurship and mass innovation" policy. Zhengzhou, the capital of Henan Province, had a GDP of 770.24 billion yuan in the first half of 2022 and a growth rate of 8.2%. In the first quarter of 2022, the province had 5.4% industrial value-added growth, going from 27th to 15th. SMEs in China are organizations that employ less than 1,000 employees or earn less than 400 million yuan in annual revenue. Data was collected using survey questionnaires. To ensure content validity, we conducted a pilot test among 30 managers working in SMEs. The reliability test was employed to identify Cronbach's alpha value (above 0.70). Before distributing the questionnaire, we conducted an ethical review. Since the questionnaire was mainly distributed in Henan Province, we conducted an ethical review at Henan University (ZDREC). According to China National Health Science and Education Development (2023) Issue no. 4, this study was exempted from ethical review because it did not cause harm to the human body and did not involve sensitive personal information or commercial interests. We nevertheless obtained an introduction letter and consent form from the Ethics Committee clearly expressing the reasons for the study and obtaining respondents' consent. When filling out the questionnaire, we gave the respondents a consent form and asked them whether they would like to participate in the study.

A total of 20 cities were included in the survey, and the initial survey phase took place between March 2022 and July 2022. We distributed questionnaires using stratified sampling, with the industry as a stratified standard, covering manufacturing, service, retail, technology, and other industries. Although this method may make the data source complex, it can improve the representativeness of the sample and the determination of the estimated value of the total quantity index. However, Agrawal (2009) demonstrates that the utilization of automated web-harvesting algorithms can easily provide the researcher with zero-cost machine-readable datasets for further analysis [135]. A total of 410 questionnaires were distributed, out of which 57 surveys were excluded due to having identical answers. After checking for missing values, outliers, and data entry errors, 330 valid surveys remained. Table 2 lists the demographic

**Table 2. Sample descriptions.**

| Demographic Characteristics | Frequency | Cumulative Percentage |
|---|---|---|
| Age | | |
| 20–25 | 61 | 18.2 |
| 26–30 | 53 | 34.5 |
| 31–35 | 76 | 57.5 |
| 36–39 | 97 | 86.9 |
| 40 and above | 43 | 100 |
| Gender | | |
| Male | 160 | 48.4 |
| Female | 150 | 100 |
| Education | | |
| Intermediate | 127 | 38.4 |
| Bachelor | 154 | 85.1 |
| Master or PhD | 49 | 100 |

information of the respondents. The study was estimated using G*power 3.1.9.7 software. G*Power was designed as a general-purpose, stand-alone power analysis program for statistical tests commonly used in social and behavioral research [128, 129, 136]. A = 0.05 (two-sided test), certainty = 1-β = 0.9, smaller effect value f2 = 0.05, and the number of independent variables = 3 were selected, thus the sample size of 288 cases was calculated. Considering a response rate of 90%, the study should include $288/0.9 \approx 320$ cases to ensure the scientific validity of the study design.

In this study, we used Analysis of Moment Structures (AMOS) 24.0 software to test the hypotheses. AMOS is typically used for confirmatory factor analysis (CFA) and structural equation modeling (SEM). AMOS is suitable for testing hypotheses or theories about relationships between variables [137]. AMOS is also suited for models where accounting for measurement errors is critical as it provides detailed diagnostics and fit indices.

## Variable measurement

A total of six principal constructs were evaluated in this study: ERM,OA, EO, efficiency-centered BMI, novelty-centered BMI, and SME performance. A 7-point scale was used to measure how much respondents agree or disagree with each survey item (1 = strongly disagree, 7 = strongly agree).

**Dependent variables.** The research employed a subjective measurement of overall business performance utilizing the paradigm of Venkatraman and Ramanujam (1986) [138]. Multiple scholars have argued that subjective performance measurements are a reliable substitute for objective performance metrics [139]. Dawes (1999) [140] argued that while utilizing an unbiased assessment of a company's overall performance offers certain benefits, companies may be hesitant to reveal their true performance statistics. Practical considerations also encompassed the gathering of empirical data within a survey of significant magnitude. This study assessed the firm's performance using eight specific criteria. These criteria included five financial performance measures (sales growth, profit growth, return on investment, net income, and market value) and three market performance measures (speed to market, market share, and penetration rate). The measurement of all items was conducted using a 7-point Likert scale, as described by Venkatraman and Ramanujam (1986).

**Independent variables.** For ERM, this approach to managing all the risks an organization faces involves integrating risk management practices into the firm's strategic, operational, and

financial activities. Key components include risk identification, risk assessment, risk response, risk monitoring, and communication. We used six items to measure ERM based on Rehman and Anwar (2019) [141], with 0.70 reliability. These items have been validated in emerging firms with satisfactory convergent validity and composite reliability. Rehman and Anwar (2019) used six items, but during the pilot test, we found that the validity of one item, namely "To ensure effective risk management, our firm have established standardized protocols for identifying both risks and opportunities within our operations," did not reach 0.7; thus, 5 items were retained. The other items are as follows:

1. Our organization has implemented a comprehensive risk management policy to address significant risks that may impact our strategic objectives.

2. A thorough analysis of risks and opportunities is the foundation for determining appropriate risk management strategies.

3. We have well-defined procedures for implementing measures to mitigate the identified risks.

4. Top management and the board of directors are regularly informed of risks.

5. The implementation of risk mitigation measures is closely monitored using standardized protocols.

OA refers to an organization's ability to rapidly adapt to market and environmental changes in productive and cost-effective ways. It encompasses flexibility, responsiveness, and the ability to innovate. We adapted six items proposed by Cegarra-Navarro et al. (2020) [142] with a high level of reliability. The items are:

1. We possess the agility to address customers' requirements promptly.

2. We demonstrate agility in swiftly adjusting our production/service delivery in response to demand fluctuations.

3. We have the capability to promptly resolve issues arising from suppliers.

4. We exhibit agility in implementing decisions to address market changes effectively.

5. Innovation and reshaping are active goals for our organization.

6. In our view, market changes represent a favorable opportunity for rapid exploitation.

EO is defined by dimensions such as innovativeness, proactiveness, and risk-taking. It reflects an organization's strategic posture towards entrepreneurship. Proactiveness, risk-taking, and innovativeness were taken as a single EO variable, with the three EO dimensions assessed as a single EO construct per Knight (1997) [76] and Covin and Slevin (1989) [42].We adapted an original survey containing 10 items. However, three of these did not apply to SMEs in China and were deleted as the validity did not reach 0.7. The remaining items are

1. My firm proactively takes action in response to competitors, prompting them to react accordingly.

2. My firm is dedicated to innovating products, services, administrative techniques, and operating technologies.

3. Leaders in our organization strongly prefer introducing novel ideas and products.

4. Our company has successfully marketed a range of new products and services over the past five years.

5. Creating a culture of innovation within the company is a key priority for top managers.

6. We consistently find our top managers capable of introducing new products and ideas ahead of the competition.

7. High-risk projects with substantial return potential are attractive to our company.

**Mediator variables.** BMI involves redefining how a company creates, delivers, and captures value, often leading to a significant shift in its operations and market position. It encompasses changes in revenue models, cost structures, value propositions, and delivery mechanisms. Based on Zott and Zmit (2008) [99], BMI is classified into efficiency-centered BMI and novelty-centered BMI. There were originally 13 questions to measure efficiency-centered BMI; however, a pilot test found that the validity of six items did not reach 0.7; therefore, six items were deleted. The following items were included:

1. We take active measures to reduce the price of our products or services.

2. Our company takes active measures to reduce search and communication costs for our partners.

3. The business model has a very low error rate in transaction execution.

4. The company takes active measures to speed up transactions.

5. Transactions are transparent, and information, products, and services can be tracked.

6. Our company places importance on providing convenient and fast service to our customers.

7. Overall, our business model offers high transaction efficiency.

Regarding novelty-centered BMI, there were originally 11 questions. However, a pilot test found that the validity of four items did not reach 0.7; therefore, four items were deleted. The following items were included:

1. In general, the company's business model is novel.

2. Our company is constantly introducing innovations to its business model.

3. Our company's business model relies on trade secrets or copyrights.

4. Our company has been awarded many patents due to the excellence of various aspects of its business model.

5. This business model connects participants to transactions in a novel way.

6. The business model offers an unprecedented variety and number of participants/goods/services.

7. We are actively taking steps to combine products, information, and services in new ways.

## Results and discussion

### Empirical strategy and measurement model

We conducted several statistical tests to analyze the data. First, we performed a CFA to verify the validity of the measurement model. In the second step, we evaluated model's goodness of fit. The measurement model consists of six variables: ERM, OA, EO, efficiency-centered BMI,

novelty-centered BMI, and SMEs' performance. We assessed the adequacy of the model by examining different fit indices, including chi-squared ($x^2$/df), root-mean-square error of approximation (RMSEA), comparative fit index (CFI), Tucker-Lewisindex (TLI). Suitable model fit indices suggest that the measurement model is satisfactory; **Table 3** provides the details for all indices. Next, we added pathways to the graphic to depict the postulated connections between latent variables. In the last step, we estimated the structural model and calculated path coefficients, standard errors, and significance levels for the hypothesized relationships.

## Validity and reliability

We measured convergent validity by examining the composite reliability and average variance extracted from the measures [143]. Skewness and kurtosis were less than 3, indicating data with a normal distribution and few outliers. According to the data in **Table 4**, the outer loadings were greater than 0.5, exceeding the acceptance standard of significance. Moreover, the results of the convergent validity test show that the composite reliability (CR) value represents a more significant factor than the 0.7. In addition, the questionnaire's average variance extracted (AVE) value is acceptable compared to the accepted standard of 0.5, suggesting that it is generally valid and convergent. According to the discriminant validity test results, indicators of discriminant validity are positive (see **Table 5**). There is a simple correlation coefficient between the variables, and the square root of their AVE values is lower than the correlation coefficient between them. Due to its validity and reliability, this scale complies with both the validation and reliability requirements.

## Hypothesis testing

This study builds the initial SEM pathway map using the conceptual model AMOS24.0, which is based on the research model proposed in **Fig 2**. The figure includes six variables: three endogenous variables, namely ERM, OA EO, and three external derivative variables, namely novelty-centered BMI, efficiency-centered BMI, and SME performance. The initial SEM was analyzed, and the goodness of fit was interpreted.

As shown in **Table 6**, the $X^2$ value of the initial model fit is 2.359 (degree of freedom DF = 636), indicating a good fit. At the same time, the TLI value and CFI value of the initial model are 0.90 and 0.935, respectively, exceeding the thresholds for significance of $> 0.9$ and $< 0.8$. The RMSEA value is 1.61, and the SRMR = 0.81. The initial model fit index shows that the initial model fits well with the sample data. The relevant test results of the initial SEM model are given in **Tables 7 and 8**. Finally, in line with our expectations, efficiency-centered BMI mediates the relationship of ERM and SME performance (indirect effect = 0.15, CI 95% = [0.06, 0.03] excluding 0), OA positively affects SME performance via efficiency-centered BMI (indirect effect = 0.32, CI 95% = [0.05, 0.59] excluding 0), and EO has a significant impact on

**Table 3. Measurement model.**

| Index | Measurement | value |
|:---:|:---:|:---:|
| $x^2$/df | --- | p < 3 |
| CFI | 0 ~ 1 | > 0.09 |
| IFI | 0 ~ 1 | > 0.09 |
| TLI | 0 ~ 1 | > 0.09 |
| SRMR | 0 ~ 1 | < 0.05 |
| RMSEA | 0 ~ 1 | < 0.05 |

**Table 4. The standard procedure is to load items by using AVE, CR, and OL.**

| Factors to consider | Item | Std. Distinction | Skewness | Kurtosis | OL | AVE | CR |
|---|---|---|---|---|---|---|---|
| F6<br>SMEs' performance | CV1 | 1.615 | -0.877 | -0.741 | .692 | .59 | .85 |
| | CV1 | 1.497 | -0.808 | -0.538 | .675 | | |
| | CV3 | 1.257 | -0.923 | -0.11 | .740 | | |
| | CP1 | 1.701 | -0.71 | -0.199 | .711 | | |
| | CI1 | 1.633 | -1.051 | 0.223 | .699 | | |
| | CI2 | 1.649 | -0.807 | -0.069 | .617 | | |
| | CI3 | 1.647 | -0.742 | 0.131 | .673 | | |
| | CI4 | 1.685 | -0.901 | -0.766 | .692 | | |
| F5<br>Novelty-centered BMI | ES1 | 1.787 | -0.961 | -1.168 | .583 | .54 | .77 |
| | AI1 | 1.923 | -0.799 | -1.07 | .770 | | |
| | AI2 | 1.846 | 0.374 | -0.914 | .781 | | |
| | AI3 | 1.717 | 0.538 | -0.887 | .590 | | |
| | AI4 | 1.733 | 0.476 | -0.817 | .764 | | |
| | AI5 | 1.74 | 0.513 | -0.91 | .756 | | |
| | AI6 | 1.592 | 0.363 | -0.766 | .666 | | |
| F4<br>Efficiency-centered BMI | ES2 | 1.73 | -1.05 | -0.326 | .695 | .56 | .80 |
| | ES3 | 1.89 | -0.678 | -0.468 | .681 | | |
| | CT1 | 2.17 | -0.989 | -0.268 | .752 | | |
| | CT2 | 1.709 | -0.081 | -0.461 | .683 | | |
| | CT3 | 1.743 | 0.396 | -0.231 | .767 | | |
| | CT4 | 1.794 | 0.395 | -0.409 | .766 | | |
| | CT5 | 1.697 | 0.432 | -0.418 | .584 | | |
| F3<br>EO | TF1 | 1.866 | 0.351 | -0.217 | .670 | .51 | .85 |
| | TF2 | 1.822 | -0.084 | -0.159 | .649 | | |
| | TF3 | 1.867 | 0.37 | -0.398 | .769 | | |
| | TF4 | 1.817 | 0.311 | -0.991 | .747 | | |
| | TF5 | 1.874 | 0.416 | -0.648 | .736 | | |
| | CF4 | 1.574 | 0.411 | -0.809 | .654 | | |
| | CF5 | 1.842 | -0.257 | -0.786 | .681 | | |
| F2<br>OA | CF2 | 1.637 | 0.301 | -0.655 | .669 | .51 | .85 |
| | SF1 | 1.81 | 0.385 | 0.219 | .664 | | |
| | SF 2 | 1.821 | 0.378 | -0.691 | .673 | | |
| | SF 3 | 1.854 | 0.438 | 0.202 | .655 | | |
| | SF 4 | 1.848 | 0.05 | 1.423 | .751 | | |
| | SF 5 | 1.856 | -0.134 | 0.807 | .650 | | |
| F1<br>EMR | BT1 | 2.159 | -0.227 | -0.977 | .646 | .51 | .85 |
| | BT 2 | 1.719 | -0.742 | -0.9 | .655 | | |
| | BT 3 | 1.848 | -0.93 | -0.918 | .769 | | |
| | BT 4 | 1.801 | -0.76 | -1.442 | .647 | | |
| | BT 5 | 1.81 | -0.883 | -0.931 | .636 | | |

SME performance via efficiency-centered BMI (indirect effect = 0.01, CI 95% = [0.0, 0.17] excluding 0). Hence, hypotheses 6a, 6b, and 6c are accepted.

## Main findings

This study examines the mediating role of BMI in the relationship between ERM, OA, EO, and SME performance. The findings support that SMEs can improve their performance by

**Table 5. Descriptive statistics and correlation matrix.**

| Variable | F1-ERM | F2-OA | F3-EO | F4-NBMI | F5-EBMI | F6-SMEsP |
|---|---|---|---|---|---|---|
| F1-ERM | 0.71 | | | | | |
| F2-OA | 0.117* | 0.64 | | | | |
| F3-EO | 0.111** | 0.095** | 0.66 | | | |
| F4-EBMI | 0.142** | 0.112** | 0.122* | 0.71 | | |
| F5-NBMI | 0.132* | 0.21** | 0.195* | 0.086* | 0.69 | |
| F6-SMEsP | 0.231* | 0.067* | 0.087** | 0.126** | 0.065* | 0.67 |
| $\sqrt{AVE}$ | 0.73 | 0.81 | 0.92 | 0.79 | 0.83 | 0.8 |

Notes: N = 330

* $p < 0.05$

** $p < 0.01$

N/A: not suitable for analysis.

enhancing their ERM, OA, EO, and efficiency-centered BMI. Novelty-centered BMI does not mediate the relationship between ERM, OA, and EO and SME performance. The summary of the hypotheses is available in **Table 9**. The reasons novelty-centered BMI does not have a positive effect on SME performance may be because, first, ERM encourages companies to conduct BMI, but not all BMI applications necessarily led to performance improvement. ERM tends to avoid risks, while novelty-centered BMI tends to embrace extensive and adventurous innovation. These two effects may offset each other, meaning that there is no significant improvement in SME performance. Second, novelty-centered BMI may not address a genuine market need or problem and is likely to fail. Insufficient understanding of market demands and customer preferences can lead to products or services that do not resonate with the target audience. Bocken and Snihur (2020) [144] found that in highly volatile environments, novelty-centered BMI can lead to misalignment with market demands, resulting in reduced performance. Another possible reason is that poor execution of innovative ideas, including ineffective project management and operational inefficiencies, can derail innovations. Unresolved technical challenges and reliance on immature or unproven technologies can lead to failure. Wirtz et al. (2016) [145] point out that novelty-centered BMI can sometimes create market confusion and dilute the brand, leading to performance declines. Madhavan et al. (2022) [146] indicate that novelty-centered BMI alone does not guarantee sustainable performance improvements without incorporating efficiency elements and adapting to environmental turbulence. Zott and Amit (2007) [4] highlight cases where novelty-centered BMI leads to overcomplexity and operational inefficiencies, thereby negatively impacting firm performance.

Conversely, efficiency-centered BMI not only directly affects SME performance, it also plays a mediator role in the relationship between ERM, OA, and EO on SME performance. ERM helps identify potential risks and inefficiencies within an organization; efficiency-centered BMI uses these insights to streamline operations, cut costs, and allocate resources more effectively, thus improving overall performance. OA requires rapid adaptation to changes, and efficiency-centered BMI ensures that these changes do not lead to unnecessary costs by maintaining operational efficiency and cost control. Additionally, EO often innovates in product and market strategies, and efficiency-centered BMI ensures that such innovations are sustainable and profitable by focusing on reducing costs and optimizing resources. These findings are consistent with those of previous studies. For example, Wei Cui et al. (2023) [147] observe that efficiency-centered BMI helps in reducing transaction costs and enhancing overall operational efficiency, which in turn supports better performance outcomes for firms engaging in green

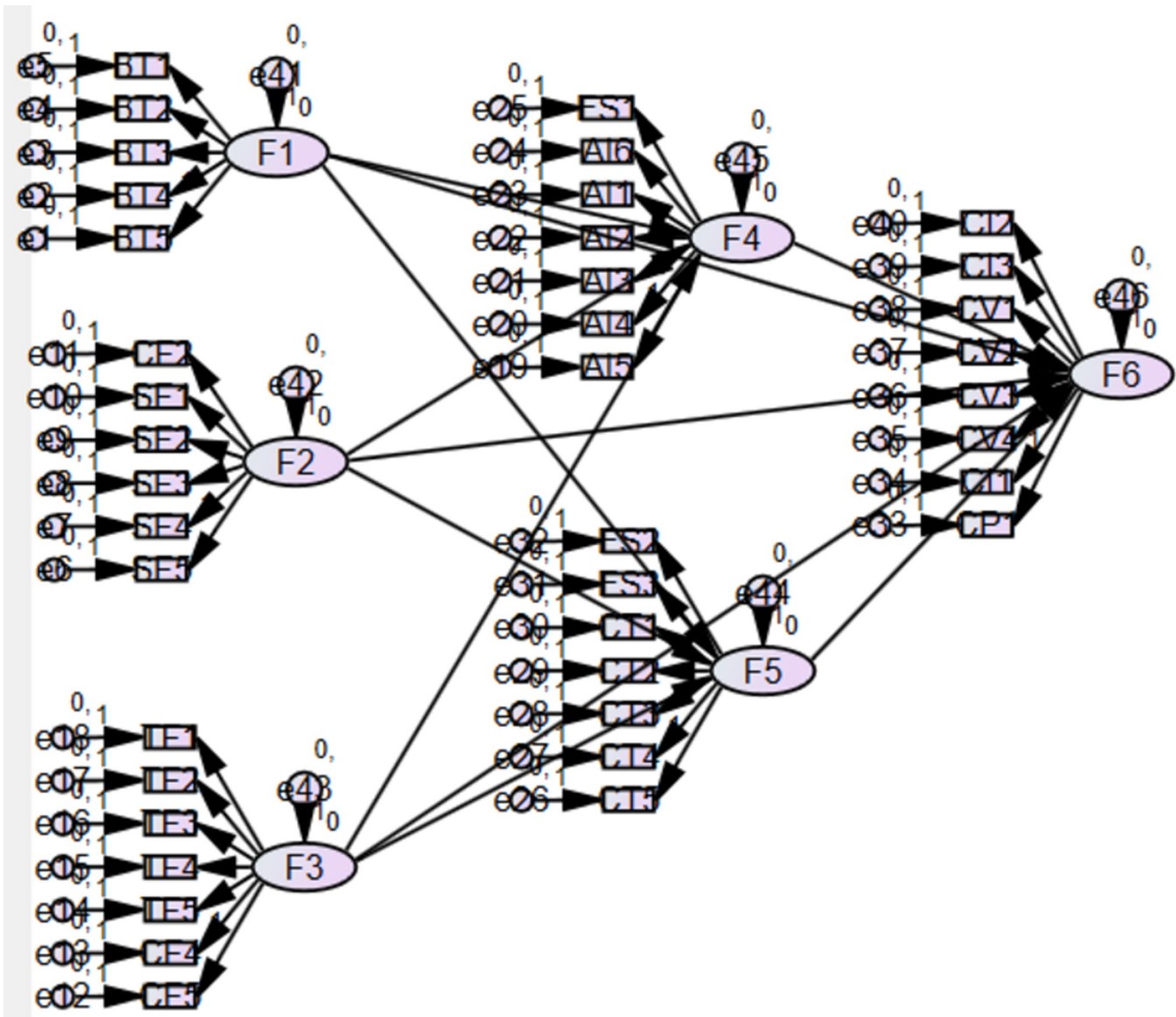

**Fig 2. Standardized estimates of independent variables BMI and SME performance.**

innovations. Loon et al. (2019) [148] reveal that high-technology SMEs in Hong Kong benefit from efficiency-centered BMI by balancing the need for innovation with operational efficiency.

The results support that BMI theoretically includes efficiency improvements to existing business models (efficiency-centered BMI) and the development of entirely new business models to explore new markets or products (novelty-centered BMI). However, in practice, especially in resource-limited SMEs, efficiency-centered BMI is often prioritized over novelty

**Table 6. Values of fit indices.**

| $x^2$/df | RMSEA | TFI | CFI | NFI | IFI |
|---|---|---|---|---|---|
| 2.359 | 1.61 | 0.90 | 0.935 | 0.921 | 0.952 |

**Table 7. Estimated results from AMOS24.0.**

|  |  |  | Estimate | S.E. | C.R. | P |
|---|---|---|---|---|---|---|
| F4 | <--- | F1 | 0.238 | 0.048 | 4.958 | *** |
| F4 | <--- | F2 | 0.221 | 0.046 | 4.767 | *** |
| F4 | <--- | F3 | 0.322 | 0.116 | 2.778 | *** |
| F5 | <--- | F1 | 0.229 | 0.05 | 4.571 | ** |
| F5 | <--- | F2 | 0.177 | 0.048 | 3.707 | *** |
| F5 | <--- | F3 | 0.338 | 0.122 | 2.765 | *** |
| F6 | <--- | F1 | 0.307 | 0.052 | 5.851 | ** |
| F6 | <--- | F2 | 0.192 | 0.048 | 4.025 | *** |
| F6 | <--- | F3 | 0.476 | 0.127 | 3.76 | *** |
| F6 | <--- | F4 | 0.125 | 0.06 | 2.083 | *** |
| F6 | <--- | F5 | 0.006 | 0.057 | 0.111 | 0.912 |

Note: * $p < 0.05$

** $p < 0.01$

*** $p < 0.001$

BMI, and there are potential contradictions between the theoretical and empirical results. In theory, novelty-centered BMI brings disruptive change and long-term competitive advantage as it usually involves the creation of entirely new value propositions and market opportunities. However, implementing novelty-centered BMI may be more challenging for SMEs due to financial, technological, and managerial experience constraints. These findings demonstrate that SMEs prioritize efficiency-centered BMI (e.g., optimizing production processes, reducing costs, or improving the productivity of existing products) because it usually requires a lower initial investment and results can be seen more quickly.

Over the past two decades, the dynamic capability view (DCV) has become one of the most active topics in strategic management. Scholars have identified several dynamic capabilities, such as organizational ambidexterity [149], resource divestiture [150], absorptive capacity [151], and product innovation [152], and have examined their impact on firm performance. However, knowledge on how and which specific dynamic capabilities influence BMI is still lacking [6]. In order to address the challenges of BMI, firms must build dynamic capabilities that are tailored to their needs and customer base [153].

Among the various dynamic capabilities that exist, this paper examines four specific dynamic capabilities, including ERM, OA, EO, and efficiency-centered BMI. These dynamic capabilities, both individually and in combination with each other, contribute to organizational success. It is worth noting that although novelty-centered BMI does not directly lead to performance improvement, efficiency-centered BMI can be seen as an innovation, especially in resource-constrained and fiercely competitive market environments. Efficiency-centered

**Table 8. Results of the mediating role of efficiency-centered BMI.**

|  |  |  | Indirect Effect | BC (95% CI) |
|---|---|---|---|---|
| Bootstrapping |  |  |  |  |
| ERM→ EBMI →SME performance |  |  | 0.15 | 0.06, 0.03 |
| OA→ EBMI →SME performance |  |  | 0.32 | 0.05, 0.59 |
| EO→ EBMI →SME performance |  |  | 0.10 | 0.01, 0.17 |

**Table 9. Hypotheses testing.**

| Hypothesis | Results |
|---|---|
| Hypothesis 1. ERM positively influences SME performance. | Supported |
| Hypothesis 2. OA positively influences SME performance. | Supported |
| Hypothesis 3. EO positively influences SME performance. | Supported |
| Hypothesis 4. Efficiency-centered BMI positively influences SME performance. | Supported |
| Hypothesis 5. Novelty-centered BMI positively influences SME performance. | Not Supported |
| Hypothesis 6 (H6a). Efficiency-centered BMI mediates the relationship between ERM and SME performance. | Supported |
| Hypothesis 6 (H6b). Efficiency-centered BMI mediates the relationship between OA and SME performance. | Supported |
| Hypothesis 6 (H6c). Efficiency-centered BMI mediates the relationship between EO and SME performance. | Supported |
| Hypothesis 6 (H6d). Novelty-centered BMI mediates the relationship between ERM and SME performance. | Not Supported |
| Hypothesis 6 (H6e). Novelty-centered BMI mediates the relationship between OA and SME performance. | Not Supported |
| Hypothesis 6 (H6f). Novelty-centered BMI mediates the relationship between EO and SME performance. | Not Supported |

BMI helps companies improve their financial performance and market position in the short term and enables them to accumulate capital and experience for future radical innovation.

More importantly, achieving performance excellence and sustainable growth is a complex evolutionary process that requires enterprises to pay attention to the changes in the external environment, manage a variety of knowledge dynamically, and—even more importantly—establish a good business model. Thus, a more integrated and systematic theory for achieving superior performance and sustainable growth is needed, especially regarding the growth path of innovative firms in the new economic environment. From the dynamic capability perspective, this study provide a more robust explanation for achieving superior performance and sustainable growth.

## Implications for practice

Practices related to ERM, OA, and EO helped SMEs develop effective BMI, which indirectly enhanced their performance. The importance of ERM, EO, and OA for SME performance was also supported. ERM, OA, and EO performance are mediated by efficiency-centered BMI, which impacts the path to firm performance. Two important implications arise from this research. First, industrial firms should focus on ERM, OA, and EO practices to improve their BMI and increase performance. Second, SMEs should explore why novelty-centered BMI does not influence their outcomes and how to improve BMI to contribute more to their growth.

To make this study more practical, we performed a case study of a SME, a home furnishing enterprise called Daxin (DX) founded in 1999, to implement our recommendations. Following the onset of COVID-19, the company experienced a consistent decrease in profitability and a loss in its market share. It has therefore become vital to enhance its commercial position through various methods.

We suggest that SMEs pay attention to ERM, which enables SMEs to identify, assess, and manage internal and external risks, protecting the firm from potential adverse impacts while also identifying and capitalizing on possible opportunities. We identified several main risks facing the furniture industry, including policy risks, economic risks, environmental risks,

social responsibility risks, and supply chain risks. We recommend paying attention to national industrial policies and seeking opportunities to build customized furniture projects. At the same time, close contact with relevant government departments is maintained to recognize policy trends promptly and ensure customized furniture projects can enter the implementation stage as soon as possible. To reduce product costs, the customized furniture project should continue to make technical improvements and management innovations and adopt energy-saving and emission-reduction production methods. Establishing solid cooperative relationships with downstream customers and forming a reliable sales network are recommended to reduce the adverse impact of market fluctuations on the customized furniture project. To cope with environmental risks, customized furniture projects should ensure that all production and operational activities comply with national and regional environmental regulations and standards. Additionally, customized furniture projects should consider the planning of environmental protection facilities at the design stage and adopt clean production technologies to reduce the negative environmental impact. Regarding social responsibility risk, customized furniture projects must actively fulfill the customized furniture industry's social responsibility obligations and show the public how enterprises fulfill their social responsibility through transparent reports. Customized furniture projects can also establish long-term partnerships with the community and participate in community building, education, environmental protection, and other public welfare undertakings to establish an optimistic customized furniture industry corporate image. To cope with supply chain risks, customized furniture projects need to establish a perfect supplier management system and solid cooperative relationships with suppliers. Adopting information technology means improving the visibility and transparency of the supply chain to better cope with supply chain issues. Businesses should establish contact with alternate suppliers to ensure that the customized furniture project can quickly switch to an alternate supply chain in case of a problem with the primary supplier.

Additionally, we suggest that DX promote BMI and implement modular manufacturing and management techniques. By inputting their extensive collection of modular data into the software, dealers can easily submit orders, transmit the data to the production department, and then categorize the output. The software decreases the amount of time it takes to complete a task and lowers expenses. Strategically, the implementation of a modular production and sales integration system will lead to the development of a competitive advantage in the market through a combination of differentiation and cost leadership. This advantage will be difficult for competitors to replicate.

After COVID-19, China has implemented a range of measures to assist high-tech and innovative SMEs, such as providing loans, offering people training, granting tax relief, and more. We therefore recommend that DX capitalize on this policy and utilize mature platforms to improve the configuration of smart homes. Shift from traditional furniture firms to technology-based SMEs through the integration and development of big data, artificial intelligence, e-commerce, and other advanced technologies. Establish a new home shopping platform using the O2O model. Create a novel industrial ecosystem that combines physical logistics, services, and customer experiences with digital business processes, financial transactions, and information exchange. Establish an intelligent and interconnected retail and service framework that spans all aspects of a household's needs.

## Research limitations and future directions

Despite this study's significant policy implications, future research may need to address several limitations. First, this study used a small sample. As SMEs in China are relatively concentrated in the central region, most of the questionnaires were distributed in Henan due to population

homogenization, and considering labor and time costs, the sample may not be able to replace the whole. However, the preliminary findings of the current research can provide valuable insights before expanding to a national sample. It also helps to improve research tools and provide more space for follow-up research. The second limitation is that the measurement of ERM, OA, EO, BMI, and SME performance is mainly based on existing research findings from authoritative domestic and international journals in related fields, meaning that the authority and reliability of the research results are guaranteed. However, it should be noted that the concepts involved in the study have yet to be studied and tested more maturely based on the Chinese scenario. All the scales used in this study are existing scales from abroad with unique cultural, time, and language limitations.

There is opportunity for research on the relationship between technology and BMI as a precursor. Our findings indicate that BMI has no positive effect on performance. Nevertheless, a company's competitive advantage is derived from its rare technology or resources that are not replicable. For instance, the ongoing innovation by Apple. On the one hand, enterprises need to design appropriate BMIs for technological innovation to promote product innovation. On the other hand, to successfully use BMI to maintain a compelling long-term competitive advantage, future research needs to first break through the barrier of technological innovation. It can monopolize the allocation of scarce resources. These competitive advantages can be constructed through technological change.

In addition, market dynamics and BMI are promising avenues for future research. Even if an enterprise's BMI is currently thriving, it needs to be improved or even subverted with the changes in the business environment and the development of the enterprise's situation. Under what circumstances does the business model of innovation need to change, and what is the evolution path? This is significant, but the research is complicated. Business models need to be innovated, and it is essential to identify the market segments to target for the design of the business model. A good BMI can scale across multiple segments, but knowing which segment to pursue first is critical, and a marketing strategy can help the BMI succeed. Most importantly, the elements of the BMI create differentiation from market competitors. While competitors can replicate many BMIs, it may take years. However, pioneering a new business model sometimes leads to competitive advantage. The shift in BMI is related to the generally accepted model of firm growth stages [154]. Thus, future research needs to focus on the issues of BMI and dynamic evolution.

## Supporting information

**S1 Questionnaire.**
(DOCX)

## Author Contributions

**Conceptualization:** Yan Jingwen, Shafie Bin Sidek.

**Data curation:** Yan Jingwen, Nitty Hirawaty Kamarulzaman.

**Methodology:** Azmawani Abd Rahman, Tong Tong.

**Software:** Shafie Bin Sidek.

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
