## [Decision Letter · Decision Letter 0]

24 Apr 2024

PONE-D-24-10141A study of the impact of dynamic capabilities on business model innovation and performance of SMEs based on Chinese enterprises' dataPLOS ONE

Dear Dr. Rahman,

Thank you for submitting your manuscript to PLOS ONE. After careful consideration, we feel that it has merit but does not fully meet PLOS ONE’s publication criteria as it currently stands. Therefore, we invite you to submit a revised version of the manuscript that addresses the points raised during the review process.

We look forward to receiving your revised manuscript.

Kind regards,

Marcelo Dionisio

Academic Editor

PLOS ONE

Journal Requirements:

3. We note that your Data Availability Statement is currently as follows: 

"All relevant data are within the manuscript and its Supporting Information files."

6. Please upload a new copy of Figure 1 as the detail is not clear. Please follow the link for more information: 

https://blogs.plos.org/plos/2019/06/looking-good-tips-for-creating-your-plos-figures-graphics/

https://blogs.plos.org/plos/2019/06/looking-good-tips-for-creating-your-plos-figures-graphics/

7. Please note that Table 4 is present, but Table 3 is missing. 

**Additional Editor Comments:**

Please carefully review our reviewers' comments as both suggested major revisions after carefully analyzing your manuscript. Although you have a very compelling subject, you are dealing with many concepts and variables and should be careful on yor approach and readability of your article. Take special attention on the development of your framework and with details of your methodology. 

Reviewers' comments:

Reviewer's Responses to Questions

**Comments to the Author**

1. Is the manuscript technically sound, and do the data support the conclusions?

Reviewer #1: Partly

Reviewer #2: Partly

2. Has the statistical analysis been performed appropriately and rigorously? 

Reviewer #1: Yes

Reviewer #2: Yes

3. Have the authors made all data underlying the findings in their manuscript fully available?

Reviewer #1: Yes

Reviewer #2: Yes

4. Is the manuscript presented in an intelligible fashion and written in standard English?

Reviewer #1: Yes

Reviewer #2: Yes

5. Review Comments to the Author

Reviewer #1: • The manuscript needs a clearer explanation of how dynamic capabilities like enterprise risk management (ERM), organizational agility (OA), and entrepreneurial orientation (EO) directly impact SME performance. Detailed clarification of these mechanisms would strengthen the conceptual framework.

• More comprehensive methodological details are necessary, especially regarding the structural equation modeling (SEM) used. Information on model fit indices, assumptions testing, and justification for the sample size and sampling method should be included to ensure methodological rigor.

• The operational definitions and measurement scales for dynamic capabilities, business model innovation, and performance require thorough examination. The manuscript should specify how these constructs were measured and provide validity and reliability statistics for the scales used.

• The assertion that growth in efficiency BMI fully mediates the relationship between dynamic capabilities and SME performance needs robust statistical testing, possibly through bootstrapping techniques to provide confidence intervals for indirect effects.

• The discussion of results could be enhanced by exploring why certain dynamic capabilities might have stronger effects on business model innovation than others, and how this varies across different types of SMEs.

• Integration of literature needs improvement to build a compelling argument for the study’s contributions. The manuscript should discuss contradictory findings from previous studies to position the current research within the broader scholarly debate.

• The practical implications section is too generic. A deeper discussion on how SME managers can leverage dynamic capabilities for business model innovation to enhance performance would be beneficial, potentially illustrated with examples or case studies.

• The limitations section should be expanded to address potential biases and the generalizability of the results. Future research directions could examine the role of external environmental factors such as market dynamics and technological changes.

• Clarify the usage of "Figure 1" for multiple diagrams to prevent confusion; each figure should be uniquely numbered.

• Standardize terminology such as "Efficiency BMI" and "Novelty BMI" throughout the document to avoid inconsistencies.

• Specify which version of AMOS was used for the analysis, as the manuscript mentions both AMOS 24.0 and AMOS 26.0, which could lead to credibility issues.

• Provide a justification or reference for the adequacy of the sample size for SEM to enhance the methodological section.

• Improve the clarity in presenting hypothesis testing results, possibly through a structured table format, to distinctly communicate supported and unsupported hypotheses.

• Update or remove outdated or irrelevant references to align the literature review with current research trends and improve the manuscript’s authority.

• Enhance the theoretical explanation for choosing specific mediators in the study to help readers understand the theoretical basis for expected influences.

• Expand the ethical considerations section to include details about the rights informed to participants, such as confidentiality and withdrawal rights, to improve the study's ethical rigor.

I recommend major revisions to address these issues before the manuscript is suitable for publication.

Reviewer #2: The study explores the impact of dynamic capabilities on business model innovation and performance in small and medium-sized enterprises (SMEs), particularly within the context of China's high-tech SMEs. Utilizing structural equation modeling (SEM) for data analysis, the research is based on a dataset from 330 Chinese high-tech SMEs. Here are some suggestions：

1.Regarding the color scheme consistency in Figure 1, it is essential that the colors assigned to 'resources,' 'strategy,' and 'performance' are uniformly aligned with their corresponding sections within the chart. Additionally, to more clearly depict the mediating effects, it is suggested that H6a and H6b be grouped together with H1a and H4, respectively. The same principle applies to H7a, H7b, H8a, and H8b.

2.Concerning the issue of sample selection, the article's choice of companies solely within Henan Province as the research sample may not be representative of the national standard. While the article justifies this by discussing the development of Henan Province, such reasoning may still lack persuasive power.

3.There is a numerical error; the correct figure should be 310. “The 132 questionnaires completed contained 132 valid responses.”

4.This section of the content could be organized using a list format with numerical bullet points to improve clarity and readability.

“The other items are：Our organization has implemented a comprehensive risk management policy to address significant risks that may impact our strategic objectives. A thorough analysis of risks and opportunities is the foundation for determining appropriate risk management strategies. We have well-defined procedures for implementing measures to mitigate identified risks. Top management and the board of directors are regularly informed of risks. The implementation of risk mitigation measures is closely monitored using ”

5.There is a discrepancy between the article's conclusions and the empirical findings. The empirical results indicate that the direct effect of F5 on F6 is not significant, whereas the conclusion drawn in the text states that the direct effect of F5 on F6 is significant. This inconsistency also leads to issues in the analysis of the empirical results regarding the mediating effect. This issue is prevalent in multiple parts of the article, such as the description in the abstract: "Despite the insignificant indirect relationship between enterprise risk management (ERM), organizational agility (OA), entrepreneurial orientation (EO), novelty-centered BMI, and SME performance, this pathway is fully mediated by growth in efficiency BMI."

6.The description of the research content is vague. For instance, in the "main findings" section, it is mentioned that "Among the largely unanswered questions is how BMI affects ERM and firm performance." However, in your article, only the impact of ERM on BMI was tested, without addressing the influence of BMI on ERM. Such statements can cause confusion for the readers.

7.There is a mismatch between the research data and the conclusions. The data used in the article pertain to high-tech industrial enterprises and do not cover financial institutions such as banks. Therefore, the descriptions in the "practical implications" section of the article are inappropriate.

8.The quantification of the "firm performance" indicator is problematic. "Firm performance" is used as the dependent variable, measured by a composite of eight indicators, all of which are financial in nature and unrelated to the environment. Consequently, it is not possible to draw conclusions related to "corporate environmental performance."

9.The basis for the classification of mediating variables is unclear. The article categorizes BMI into "efficiency-oriented" and "innovation-oriented" types, but it does not provide a clear explanation of how this classification is determined.

10.Limitations and Future Research Directions. The article should provide a more detailed discussion of its research limitations, including the constraints of the data collection method, the limitations of the sample size, and other factors that may influence the results. It is recommended that the authors propose directions for future research, such as examining business model innovation in different cultural and economic contexts, or exploring how various types of dynamic capabilities affect the performance of SMEs across different industries.

11.Data Accessibility. In accordance with PLOS ONE's policy, research data should be publicly accessible. The authors are requested to ensure that all relevant data are available in an appropriate public repository and to provide access information.

6. PLOS authors have the option to publish the peer review history of their article (what does this mean?). If published, this will include your full peer review and any attached files.

Reviewer #1: No

Reviewer #2: No

---

## [Author Response · Author response to Decision Letter 0]

5 Jun 2024

We would like to thank you for your valuable comments and helpful suggestions. We have studied these comments and suggestions carefully and made revisions to improve the quality of our manuscript. Some inappropriate expressions have been rewritten, outdated references have been removed, and the literature review section has been polished. The methodology section carefully explains the sampling method, sampling frame, sampling population, and pilot test explain the reliability and validity of the questionnaire used. In the results section, the consistency and inconsistency between the research and previous studies are explained, and the reasons for such results are pointed out. In the practical application section, case study is added to better explain the suggestions and measures of this research.All revisions made to the manuscript were marked yellow. We hope the revised manuscript is satisfactory. Please let us know if you have concerns or questions about response to the referees.

---

## [Decision Letter · Decision Letter 1]

21 Jun 2024

PONE-D-24-10141R1A study of the impact of dynamic capabilities on business model innovation and performance of SMEs based on Chinese enterprises' dataPLOS ONE

Dear Dr. Rahman,

Thank you for submitting your manuscript to PLOS ONE. After careful consideration, we feel that it has merit but does not fully meet PLOS ONE’s publication criteria as it currently stands. Therefore, we invite you to submit a revised version of the manuscript that addresses the points raised during the review process.

We look forward to receiving your revised manuscript.

Kind regards,

Marcelo Dionisio

Academic Editor

PLOS ONE

Journal Requirements:

**Additional Editor Comments:**

Although you have had an approved decision, I strongly suggest you to follow reviewer's 4 comments and some of reviewer's 5 suggestions as well to improve the quality of your manuscript before we find it ready for approval.

Reviewers' comments:

Reviewer's Responses to Questions

**Comments to the Author**

1. If the authors have adequately addressed your comments raised in a previous round of review and you feel that this manuscript is now acceptable for publication, you may indicate that here to bypass the “Comments to the Author” section, enter your conflict of interest statement in the “Confidential to Editor” section, and submit your "Accept" recommendation.

Reviewer #3: (No Response)

Reviewer #4: (No Response)

Reviewer #5: (No Response)

2. Is the manuscript technically sound, and do the data support the conclusions?

Reviewer #3: Yes

Reviewer #4: Yes

Reviewer #5: No

3. Has the statistical analysis been performed appropriately and rigorously? 

Reviewer #3: Yes

Reviewer #4: Yes

Reviewer #5: No

4. Have the authors made all data underlying the findings in their manuscript fully available?

Reviewer #3: Yes

Reviewer #4: Yes

Reviewer #5: (No Response)

5. Is the manuscript presented in an intelligible fashion and written in standard English?

Reviewer #3: Yes

Reviewer #4: Yes

Reviewer #5: No

6. Review Comments to the Author

Reviewer #3: (No Response)

Reviewer #4: A well produced and exhaustive paper. It's almost there, thanks for incorporating previous reviewer comments. Please do not see any of the following as a critique, given that the paper is now at my desk and I have put in substantial amounts of time to review this elaborate study, please incorporate the suggestions to crystallize the paper and also increase the visibility of the paper.

1. Kindly replace many instances of the word 'cantered' with "centered"

2. I think this may be your most significant result, if possible highlight it a bit more.

EO -> Efficiency-centered BMI -> SME Performance

Indirect effect = 0.10

95% CI = [0.01, 0.17] excludes zero

Since the 95% confidence intervals exclude zero in all three cases, it indicates the mediating role of efficiency-centered BMI is statistically significant. So while efficiency-centered BMI did not have a strong direct effect on performance, the numerical evidence supports its significant indirect/mediating role in channeling the positive effects of ERM, OA and EO on SME performance in this study. Good result.

3. You also show that while novelty-centered BMI had a positive direct effect on SME performance (path coefficient 0.125), it did not significantly mediate the relationships between ERM, OA, EO and SME performance based on the insignificant indirect effects. To me, this suggests that while novelty-centered innovations may have some benefits, they did not effectively translate the impacts of risk management, agility and entrepreneurial capabilities into better SME performance in this study's context.

4. I found two contradictions in the paper:

A. There is a contradiction between the positive direct effect of novelty-centered BMI on SME performance (path coefficient 0.125, p<0.001), and the lack of a mediating effect between novelty-centered BMI and the relationships of ERM, OA, EO with SME performance.

B. The hypotheses propose that both efficiency-centered and novelty-centered BMI will positively influence SME performance (H4 and H5). But the results only support H4 for efficiency-centered BMI, contradicting H5? Please elaborate.

C. The literature review section highlights studies indicating ERM can uncover opportunities for BMI and innovation. However, the results show ERM only relates to efficiency-centered BMI, not novelty-centered BMI for SMEs.

There is a theoretical contradiction in using the dynamic capabilities view, which emphasizes continuous innovation and change, while the results favor efficiency-centered BMI over more novel, radical innovations.

The paper argues SMEs may have limited resources to experiment with BMI, yet also cites examples of large companies like Apple and Disney successfully implementing novelty-centered BMI.

The case study recommendation for the SME Daxin emphasizes efficiency BMI through modular manufacturing, contradicting the proposal for a novel "smart home" business model leveraging entrepreneurial orientation.

Please justify (briefly) these potential contradictions in the theoretical framing versus the empirical results favoring efficiency over novelty for SME BMI.

5. Your data sources are complex, it may be difficult for researcher students to provide such data. At line # 657 or # 589, you can mention an early stage AI knowledge paper: "However, Agrrawal (Managerial Finance, 2009) shows that the utilization of automated web-harvesting algorithms can easily provide the researcher with zero-cost machine readable datasets for further analysis.

6. A brief link between CSR and SME can be mentioned : Stoian, M. C., & Plakoyiannaki, E. (2023). Understanding international CSR in SMEs. Journal of Small Business Management.

A very comprehensive paper, nicely developed. Thank you.

Reviewer #5: Review of Manuscript #: PONE-D-24-10141R1, “A study of the impact of dynamic capabilities on business model innovation and performance of SMEs based on Chinese enterprises”

In short, I am sorry that I cannot be positive at this stage of the research project. Currently, the research is at best at the exploratory phase. This can be a part of a paper in the future. Unfortunately, it is poorly written and was very to grasp the essential points and arguments. Some of the most critical sentences made no sense at all due to either typos or poor command of the language. As of now, not much can be learned from this. There is significant major additional work for it to meet publication standards.

It was not clear if H1 to H5 are replications of other papers? Perhaps, then the authors can think about a cross-cultural nuance to this project and see how these results map against studies in western countries. It is more akin to a white paper for a regional economic development project.

Most of all, I beseech the authors to have it read by peers internationally and have it corrected by a professional editor before submitting elsewhere. My apologies for this somewhat unpropitious message.

7. PLOS authors have the option to publish the peer review history of their article (what does this mean?). If published, this will include your full peer review and any attached files.

Reviewer #3: No

Reviewer #4: No

Reviewer #5: No

---

## [Author Response · Author response to Decision Letter 1]

30 Jun 2024

We sincerely thank you for your feedback, we accepted all the comments from Reviewer 4, and the specific response can be found in the file named respond to reviewers. Regarding the issue of poor writing in Reviewer 5's comments, we have changed the structure of some sentences to ensure the readability for this manuscript.

---

## [Decision Letter · Decision Letter 2]

18 Jul 2024

PONE-D-24-10141R2A study of the impact of dynamic capabilities on business model innovation and performance of SMEs based on Chinese enterprises' dataPLOS ONE

Dear Dr. Rahman,

Thank you for submitting your manuscript to PLOS ONE. After careful consideration, we feel that it has merit but does not fully meet PLOS ONE’s publication criteria as it currently stands. Therefore, we invite you to submit a revised version of the manuscript that addresses the points raised during the review process.

We look forward to receiving your revised manuscript.

Kind regards,

Marcelo Dionisio

Academic Editor

PLOS ONE

Journal Requirements:

**Additional Editor Comments:**

I kindly ask you to perform one more round of minor revision, with focus on writing, for which I strongly suggest you to perform a professional English review to improve the quality and readability of your manuscript.

Reviewers' comments:

Reviewer's Responses to Questions

**Comments to the Author**

1. If the authors have adequately addressed your comments raised in a previous round of review and you feel that this manuscript is now acceptable for publication, you may indicate that here to bypass the “Comments to the Author” section, enter your conflict of interest statement in the “Confidential to Editor” section, and submit your "Accept" recommendation.

Reviewer #4: All comments have been addressed

Reviewer #5: (No Response)

2. Is the manuscript technically sound, and do the data support the conclusions?

Reviewer #4: Yes

Reviewer #5: Yes

3. Has the statistical analysis been performed appropriately and rigorously? 

Reviewer #4: Yes

Reviewer #5: Yes

4. Have the authors made all data underlying the findings in their manuscript fully available?

Reviewer #4: Yes

Reviewer #5: (No Response)

5. Is the manuscript presented in an intelligible fashion and written in standard English?

Reviewer #4: Yes

Reviewer #5: Yes

6. Review Comments to the Author

Reviewer #4: You've done a superb job addressing some very complex points from the previous revision. "An irreversible change to a company's business model is considered Business Model Innovation (BMI), however, BMI often associated with high risk, uncertainty, and ambiguity. The effectiveness of BMI on SMEs' performance is examined by structural equation modeling (SEM)..."

This will be a highly visible paper.

Reviewer #5: (No Response)

7. PLOS authors have the option to publish the peer review history of their article (what does this mean?). If published, this will include your full peer review and any attached files.

Reviewer #4: No

Reviewer #5: No

---

## [Author Response · Author response to Decision Letter 2]

3 Sep 2024

We would like to thank you for your valuable comments and helpful suggestions. We have studied these comments and suggestions carefully and made revisions to improve the quality of our manuscript. All revisions made to the manuscript were marked BLUE. Please let us know if you have concerns or questions about response to the referees. We have performed SCRIBBR for proofreading, and certificate is attached.

---

## [Editor Report · Decision Letter 3]

8 Sep 2024

A Study of Chinese Enterprises' Business Models to Determine the Impact of Dynamic Capabilities on Innovation and Performance.

PONE-D-24-10141R3

Dear Dr. Rahman,

We’re pleased to inform you that your manuscript has been judged scientifically suitable for publication and will be formally accepted for publication once it meets all outstanding technical requirements.

Kind regards,

Marcelo Dionisio

Academic Editor

PLOS ONE
---

## [Editor Report · Acceptance letter]

27 Sep 2024

PONE-D-24-10141R3 

PLOS ONE

Dear Dr. Rahman, 

I'm pleased to inform you that your manuscript has been deemed suitable for publication in PLOS ONE. Congratulations! Your manuscript is now being handed over to our production team.

Kind regards, 

on behalf of

Professor Marcelo Dionisio 

Academic Editor

PLOS ONE